# Attitudes of Kazakh Rural Households towards Joining and Creating Cooperatives

**Samal Kaliyeva [1],*, Francisco Jose Areal [2] and Yiorgos Gadanakis [1]**

[1]   School of Agriculture Policy and Development, Agriculture Building, University of Reading,
      Reading RG6 6AR, UK; g.gadanakis@reading.ac.uk
[2]   Centre for Rural Economy, School of Natural and Environmental Sciences, Newcastle University,
      Agriculture Building, King's Road, Newcastle upon Tyne NE1 7RU, UK;
      francisco.areal-borrego@newcastle.ac.uk
*   Correspondence: s.kaliyeva@pgr.reading.ac.uk

**Abstract:** The government of Kazakhstan is currently developing strategies and policies to stimulate milk production at an industrial production level to increase milk processing capacity. We use and expand the reasoned action approach as a framework to study the factors underlying the rural household's motivation to participate in a governmental programme aimed at increasing rural cooperative production in Kazakhstan to increase milk production using primary data acquired from 181 randomly selected dairy households in the Akmola region of Kazakhstan. We account for the rural household's psychological factors and socio-demographic characteristics along with the household's risk attitudes, production structure, level of information about the government support programme and cooperatives, cultural aspects as well as the household's proximity to the main market. A bivariate probit model is used to jointly estimate the impact of these factors on the rural household's intention to join and create a cooperative. The results show that rural households which hold positive views towards cooperatives, have a relatively high production capacity, are aware/know of cooperatives, and do not have a dairy business as a source of household income are relatively keen to participate in collective actions. Perceived social norms and household risk attitudes also play a significant role in the rural household's intention to participate in collective actions. Finally, gender and nationality are found to be positively associated with joining and creating a cooperative, while higher educated rural households are found to be less motivated to participate in the programme. In order to stimulate milk production at an industrial production level through a policy that encourages collective action, we recommend a policy that (a) supports rural households which have the capacity to produce and are in need; (b) is attractive to rural households which consider dairy as a source of income; and (c) is well disseminated and well explained to the targeted rural households.

**Keywords:** cooperative creation policy; dairy households; reasoned action approach; bivariate probit model; Kazakhstan

---

## 1. Introduction

During the Soviet Union (SU) regime, agricultural production in Kazakhstan was carried out through state owned sovkhozes and collective farming kolkhozes [1–3]. After the collapse of the USSR, Kazakhstan faced the problem of transitioning from a communist-collective economy to a private property-based economy [4]. The majority of kolkhozes were disintegrated, and all their former members were given shares of the holdings, proportionate to their property rights [1]. This had implications on agricultural production, including livestock. As the number of cattle and cows owned by enterprises fell sharply, the number of cattle and cows owned per household and small/peasant

farms increased steadily (Data was derived from the official website (https://stat.gov.kz) of the Statistics Committee of the Republic of Kazakhstan). Thus, the dismantling of collective farms led to a restructuring of the farming sector with three types of farms emerging: (a) agricultural enterprises (generally previous sovkhozes and kolkhozes) [1,2,4], (b) small/peasant farms ("*a joint family labour union in which individual entrepreneurial activities are directly linked with the use of land for agricultural purposes to produce, process and market farm outputs*" [5], where the number of livestock has a range between ten and thousand [6]), and (c) rural households (Households, formally known as personal subsidiary farming, tend to have small homestead land with an average area of 0.15 hectares and the number of cows from one to ten (an average 3 per household) in each yard [7]. According to the official classification of farms, a personal subsidiary farm is understood as a type of activity for the satisfaction of own needs on a land plot located in rural and suburban areas [8]. They tend to "*operate at a small scale and keep small numbers of livestock*" [6].). Out of the three types, rural households hold the largest share of livestock production in the country. For instance, in 2018, 55% of slaughtered livestock and poultry (in slaughter weight) were produced by rural households, while for the small/peasant farms and the agricultural enterprises, this accounted for 19.3% and 25.6%, respectively. Regarding the production of milk, figures show an even more significant role by rural households—with 73.5% of the milk being produced by rural households and 19.7% and 6.8% being produced by small/peasant and agricultural enterprises, respectively.

More importantly, rural households are relatively less efficient than small/peasant farms leading to milk production in Kazakhstan being characterized by low productivity [9]. Moreover, the supply of processed dairy products from the dairy industry is lower than the actual market demand [10]. This is associated with dairy factories having a deficit of milk processing due to rural households not utilising the direct supply chain for milk to the dairy industry. It is common for rural households to trade dairy products through the use of informal trade (i.e., direct sales to consumers). In other words, "modern diaries cannot obtain sufficient milk of adequate quality for their processing operations, while lower-quality milk of rural households continues to find buyers" [11]. This informal trade is economically beneficial for rural households since they receive a higher price from directly selling to consumers and relatively low transaction costs associated with milk distribution [1]. In addition, in many rural areas of Kazakhstan, there is no formal supply chain pathway for rural households to reach dairy factories.

In order to tackle the relatively low productivity of the dairy sector, the Kazakh government considered measures to increase the milk production of rural households via the creation of co-operatives (i.e., a formal network of producers) which is expected to also facilitate the access of rural households to the supply chain through dairy processing units (The current situation is that the individuals do not own the means of production and share the means of production to produce an output, which is the case of the former Soviet Union. Through cooperatives, they will have access to technologies (not own). Access for equipment, feeding, and subsidies.)

Under the current programme—'The development of Agro-industrial complex for 2017–2021'—the government initially attempted to reduce the number of agricultural activities per household and expand agricultural production in enterprises through the creation of cooperatives in rural areas. Namely, one of the goals of the governmental plan focused on attracting rural households to cooperative production, thereby increasing the processing of dairy products by the agricultural enterprises (since dairy companies face milk scarcity and in turn, have to use their capacity only by 20–60%) (The conceptual framework of the Programme, retrieved from www.primeminister.kz). The government's plan (The governmental program downloaded from the official site of the Ministry of Agriculture RK in 2017 was reissued in July 2018 with some corrections in it, most of part about cooperatives was deleted. The new document was retrieved from http://adilet.zan.kz/kaz/docs/P1800000423) was to turn over 500,000 rural households into cooperatives within a 5-year period and receive 500,000 t of milk from cooperatives in 2021. The initial government plan was revised in July 2018 and it is no longer aimed at creating more cooperatives under the programme. The reason for such U-turn in the policy is unclear

but it may be due to the realisation that the aims were too ambitious given the resources (e.g., it might be that the budget pledged to support the creation of cooperatives was not enough to financially support rural households). Nevertheless, the idea of creating cooperatives is still relevant and it has been included in the Strategic Plan of non-commercial organization "Atameken" for 2018–2023 (The National Chamber of Entrepreneurs of the Republic of Kazakhstan, which was created in 2013 by a joint decision of the government and NEPK "Union "Atameken", https://atameken.kz/ru/pages/39-missiya-palaty). In 2019, the number of rural households involved in cooperative production was 27,200, whereas the production of cow's milk by cooperatives was 65,400 t (the country's total production was 5820,100 t of milk in the same year).

We investigate rural households' intention to create and/or join a cooperative in Kazakhstan in order to identify the key aspects that underpin their decision to participate in the governmental policy aimed at participating in rural cooperative production. Whereas the literature on cooperatives has focused on the organisation and management of the cooperatives [12–15], less focus has been put on understanding the determinants of rural households' motivation to create and/or join a cooperative [16,17]. We contribute to the latter literature in four ways: (1) by being the first paper, to our knowledge, that has used the reasoned action approach [18] to jointly explain attitudes towards joining and creating farming cooperatives; (2) expanding this approach by incorporating household's cultural features, risk attitudes, the production structure, and the level of information into the framework; (3) by using a bivariate probit model to jointly analyse rural households' intention to join/create a cooperative in the context of a transitioning economy country, Kazakhstan; (4) by supporting policy adoption through gaining understanding of how households can be motivated to join/create a cooperative.

## 2. Materials and Methods

A wide number of theories have been developed to explain an individual's behaviour [19–26]. These theories put emphasis on the fact that an individual's intentions and behaviour may depend not only on individual's demographic characteristics but also on their psychological characteristics. These include an individual's own views of what the behaviour outcome would be; barriers/difficulties, including current habits, to behave in a given way, as well as the influence of other's on one's decision process (e.g., friends, family). In our case, we investigate how a rural household's decision to participate in collective action may be influenced by both a rural household's socio-demographic characteristics (e.g., age and gender) and psychological factors (e.g., individual's positive beliefs about having guaranteed sales). We expand this framework by including cultural features, risk attitudes, the production structure, and the level of information.

In order to incorporate psychological factors influencing the rural households' decisions to participate in the governmental programme either by joining or creating a cooperative, the reasoned action approach (RAA) is utilised. Currently, RAA is widely used to explain human behaviour in different fields as well as in agricultural research [27–33].

We expand the RAA by incorporating the role of other factors, such as trust, individualistic/collectivistic behaviour, views about past regime, referred to as "cultural features", as well as risk attitudes into the framework and analysis. These factors have been previously found to influence cooperative behaviour [34–39]. Moreover, we incorporated production characteristics and policy awareness and understanding into the framework. Sultana et al. [40] highlighted the role of production features, including herd size and support (e.g., training and credit services) on the decision to function as cooperative farmers or non-cooperative farmers. Rural household's awareness of the current government policy and their understanding of the essence of cooperative production are relevant in this study [41,42]. Therefore, we incorporate all these to expand the RAA. Thus, the role of own attitudes (A), social norms (SN), and perceived behavioural control (PBC) is investigated along with cultural factors (e.g., level of trust in different groups and attitudes to the Soviet Union regime), risk attitudes, household's production capacity, awareness of the current agricultural support policy, and location (distance to major market) on the rural household's decision to join or create a cooperative.

### 2.1. Psychological Factors: Reasoned Action Approach

Under the RAA, the respondent constructs a specific attitude (A) and social norms (SN) about a behaviour, and then weighs the relative importance of them in order to perform, or not to perform, the given behaviour (Figure 1).

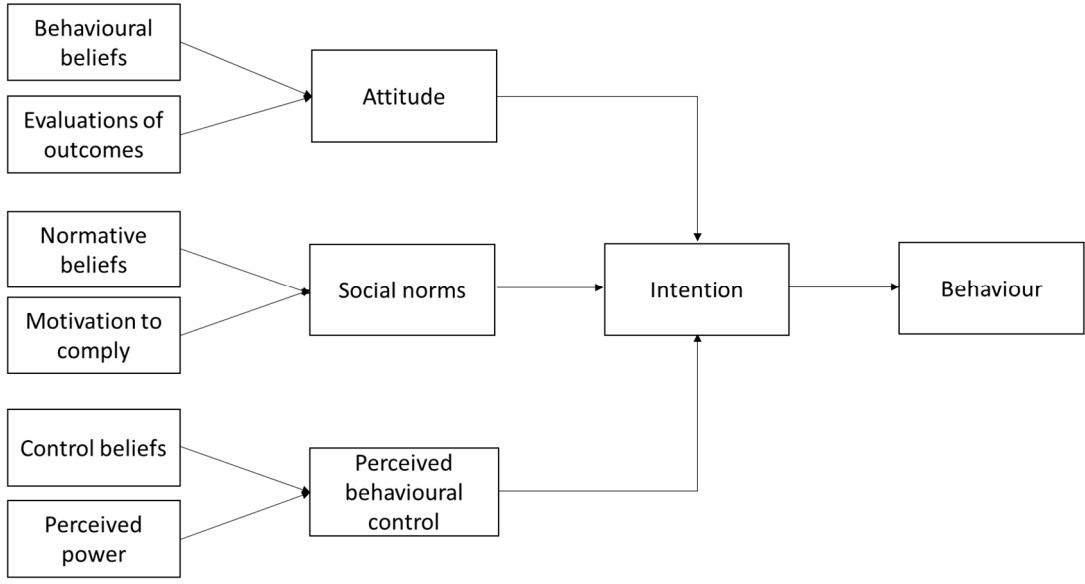

**Figure 1.** The reasoned action approach. Source [18].

Thus, following the RAA a number of statements were used for rural households to evaluate (Figure 1 and Table 1). These were used to construct indicators for attitude, social norms, and perceived behavioural control as follows:

$$A = \sum b_i e_i, \tag{1}$$

where $A$ is an individual's attitude towards a behaviour (e.g., joining a cooperative; creating a cooperative), $b_i$ is strength of belief $i$ about a consequence of a behaviour (e.g., joining a cooperative; creating a cooperative), and $e_i$ is evaluation of belief $i$. In addition:

$$SN = \sum n_i m_i, \tag{2}$$

where $SN$ is an individual's social norms towards a behaviour (e.g., joining a cooperative; creating a cooperative), $n_i$ is strength of normative belief $i$, and $m_i$ is motivation to comply with a specific normative referent referred to by $i$.

Perceived behavioural control (PBC) has always been explained as a factor which can influence on behaviour directly as well as indirectly through an intention [24]. Nevertheless, under the RAA (Figure 1), PBC consists of two components: control beliefs, which are weighted by the power of control factors [18]. Hence, it is defined as:

$$PBC = \sum c_i p_i, \tag{3}$$

where $c_i$ is the belief that control factor $i$ will be present and $p_i$ is the power of factor $i$ to facilitate or impede performance of a behaviour" (e.g., joining a cooperative; creating a cooperative) [18].

**Table 1.** Statements to reveal respondent's att=1itude, social norms, and perceived behavioural control towards the behaviour.

| Item | Questionnaire Statements | Scale |
|---|---|---|
| | Attitude | |
| B1 | Being a member of a cooperative in my region would give (gives) me a guarantee of sales | extremely unlikely–extremely likely |
| E1 | For me having guaranteed sales is | extremely bad–extremely good |
| B2 | Being a member of a cooperative in my region would give (gives) me support from recognized bodies such as Atameken, Damu and other cooperatives | extremely unlikely–extremely likely |
| E2 | For me receiving support from recognized bodies and other cooperatives is | extremely bad–extremely good |
| B3 | Being a member of a cooperative in my region would allow (allows) me to receive subsidies | extremely unlikely–extremely likely |
| E3 | For me receiving subsidies is | extremely bad–extremely good |
| B4 | Being a member of a cooperative in my region would allow (allows) me to sell a litre of milk more expensive than now | extremely unlikely–extremely likely |
| E4 | For me an increase in the price for a litre of milk is | extremely bad–extremely good |
| B5 | Being a member of a cooperative in my region would give (gives) me an ability to do business with close relatives/friends and people who I trust | extremely unlikely–extremely likely |
| E5 | For me having an ability to do business with close relatives/friends and people who I trust is | extremely bad–extremely good |
| B6 | Being a member of a cooperative in my region would require (requires) me to take responsibility for others (other members of the cooperative) | extremely unlikely–extremely likely |
| E6 | For me taking responsibility for others (other members of the cooperative) is | extremely bad–extremely good |
| | Social Norms | |
| N1 | My spouse/partner thinks that it would be (is) good for me to be a member of a cooperative in my region | extremely unlikely–extremely likely |
| M1 | With regards being a member of a cooperative in my region, I want to do what my spouse or partner thinks I should do | strongly disagree–strongly agree |
| N2 | My parents think that it would be (is) good for me to be a member of a cooperative in my region | extremely unlikely–extremely likely |
| M2 | With regards to being a member of a cooperative in my region, I want to do what my parents think I should do | strongly disagree–strongly agree |
| N3 | My best friend thinks that it would be (is) good for me to be a member of a cooperative in my region | extremely unlikely–extremely likely |
| M3 | With regards to being a member of a cooperative in my region, I want to do what my best friend thinks I should do | strongly disagree–strongly agree |
| N4 | My relatives think that it would be (is) good for me to be a member of a cooperative in my region | extremely unlikely–extremely likely |
| M4 | With regards to being a member of a cooperative in my region, I want to do what my relatives think I should do | strongly disagree–strongly agree |
| N5 | I know someone who is a member of a cooperative in my region | definitely false–definitely true |
| M5 | When it comes to matters of being a member of a cooperative, I want to be like my acquaintance | strongly disagree–strongly agree |
| | Perceived Behavioural Control | |
| C1 | I have (had) enough money to be a member of a cooperative in my region | extremely unlikely–extremely likely |
| P1 | Having enough money would make it (made it) easier for me to be a member of a cooperative in my region | strongly disagree–strongly agree |
| C2 | Being a member of a cooperative would make (makes) me dependent on decisions taken by others | extremely unlikely–extremely likely |
| P2 | Being dependent on the decisions taken by others would make (makes) it difficult for me to be a member of a cooperative | strongly disagree–strongly agree |
| C3 | I easily would find (found) like-minded people to encourage me to be a member of a cooperative in my region | extremely unlikely–extremely likely |
| P3 | Easily finding like-minded people to encourage me would make (makes) it easier for me to be a member of a cooperative in my region | strongly disagree–strongly agree |
| C4 | I easily would earn (earned) trust among fellow villagers to be a member of a cooperative in my region | extremely unlikely–extremely likely |
| P4 | Easily earning trust among fellow villagers would make (makes) it easier for me to be a member of a cooperative in my region | strongly disagree–strongly agree |
| C5 | I would have (have) different issues, including financial as a member of a cooperative in my region | extremely unlikely–extremely likely |
| P5 | Having different issues, including financial would make (makes) it difficult for me to be a member of a cooperative in my region | strongly disagree–strongly agree |

Based on Ajzen's and Fishbein's [18] argument that attitude should be about behaviour rather than about an object, we distinguish between two types of behaviour: to join an existing cooperative and to create a new cooperative. Moreover, following the RAA, each behaviour consists of four elements—action, target, time, and context. Hence, we define behaviour as 'being a member of a cooperative in my region', where:

Action: being a member,

Target: of a cooperative,

Context: in my region,

Time: unspecified, left in general.

Thus, following RAA, constructed and weighted attitudes (A), social norms (SN), and perceived behavioural control (PBC) are combined to formulate behavioural intention (BI). Table 1 shows statements used to reveal rural household's A, SN, and PBC. Respondents were asked to rate the statements on a set of unipolar and bipolar evaluative adjective scales, with five places. Following [18], to elicit attitude (A) toward being a member of a cooperative. For instance, respondents were asked to evaluate the strength of belief (B1 to B6) about a consequence of a behaviour from extremely unlikely to extremely likely (1 to 5), while respondents' evaluation of the belief (E1 to E6) were assessed from extremely bad to extremely good (−2 to +2). We translate their evaluation into scores by using Equation (1). Thus, the higher the sign of behavioural belief, the more it is expected to have a positive influence on attitude. Consequently, using Equation (1) above to sum across all scales for attitude (A), we obtain a measure of rural household's attitude towards cooperative production. Since there are 6 behavioural outcomes, the possible range of the scale for A is from −60 to +60. The same procedure was applied to reveal SN and PBC with some differences on scoring, namely, (a) respondent's normative beliefs were scored from −2 to 2 (i.e., extremely unlikely–extremely likely), while the motivation to comply with a referent was taken values from 1 to 5; (b) control beliefs were scored from 1 to 5, while the power of the factor was scored from −2 to +2 on statements capturing facilitating factors (i.e., P1; P3; P4) and from 2 to −2 on statements capturing impeding factors (i.e., P2 and P5) [18]. Hence, the range of the scale for the SN and for the PBC is (−50 to +50).

*2.2. Additional Constituents of the Model*

We incorporate to our conceptual framework with other aspects highlighted in the literature as determinants of behaviour that might also be relevant in explaining the rural household's intention to join or create a cooperative, on top of those highlighted in the RAA. These include cultural features, risk attitudes, production structure, and the level of information contained in the governmental programme.

We integrate the rural household's views about the past regime into the analysis, in order to investigate whether the rural household's intentions to become a member of a cooperative or creating one are associated with the past (i.e., whether despite the fact that almost 30 years have passed since the collapse of collective farms (kolkhozes), people's intentions to join cooperatives in Kazakhstan are related to the country's governance history). Moreover, for a post-soviet country, that has experienced the collective economy before, achievement of the aims of the governmental programme (creation of cooperative structure to support dairy production) might face several issues, such as potential mistrust by agricultural producers towards newly created structures [37]. The importance of determining the level of trust has been emphasised in previous research [15,43]. We additionally incorporate rural household's individualistic and collectivistic behaviour towards culture as a dimension as this aspect of the culture has also been emphasized in previous research [44–46]. Under 'cultural aspects', we grouped rural household's views about the past regime; the level of trust; and, their individualistic and collectivistic behaviour (Table A1, in Appendix A).

The full consequences of joining or creating a cooperative are uncertain [39]. Therefore, it is expected that rural household's decision to join or create a cooperative depends on their risk attitudes, with risk averse rural households being less likely to make changes to the status-quo. It is important

to note that cooperative membership gives access to input and output markets, which implies a risk reduction on the part of the members. We are interested in understanding whether individuals' attitudes towards risk may influence their decision to participate in collective action (i.e., join/create a cooperative).

The rural household's production structure (e.g., the potential production given the resources and the business orientation) is also considered as a potential determinant of the intention to participate in the governmental programme [36,40]. It is expected that structures currently dedicated to selling products to the market with clear business orientation may be less likely to join or create a cooperative, whereas those rural households that may need some kind of support (e.g., loan, subsidy) may be more keen on joining/creating a cooperative.

Being aware of the cooperative production policy might also have an influence on rural household's decision making. Therefore, we assume that the greater the awareness of rural households, the more likely they are to join/create a cooperative. Finally, since there may be unobserved circumstances at the district level (e.g., distance to capital, level of access to networks, district economic activity level, socio-demographic differences) that may influence individual's decisions, we have added dummy variables for each selected rural district. Consequently, Figure 2 shows the conceptual framework used in this paper.

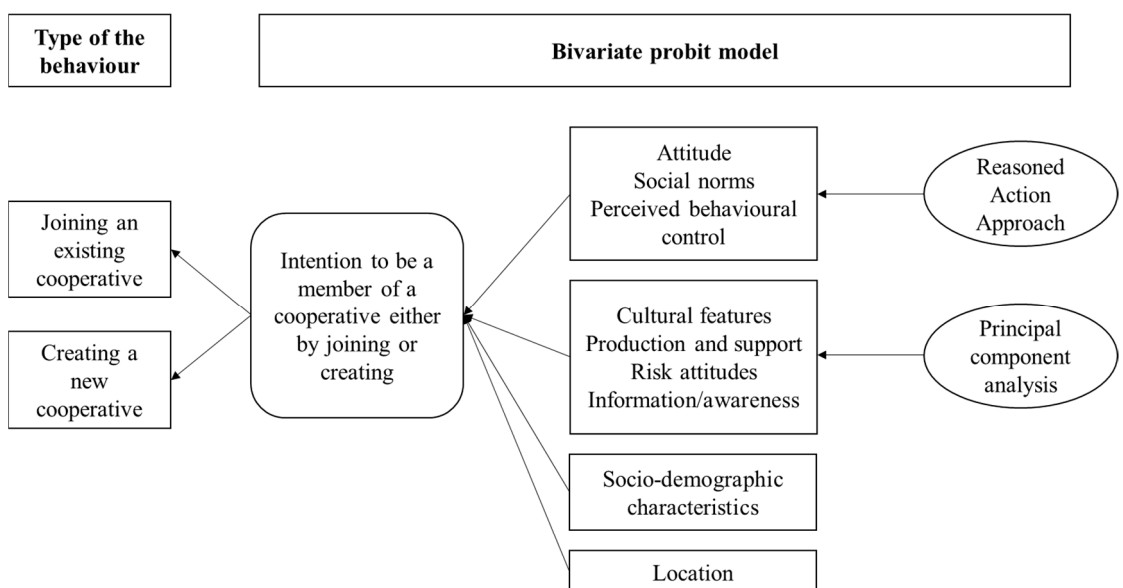

**Figure 2.** The conceptual framework for defining the factors underlying the behaviour.

### 2.3. Survey and Questionnaire

We conducted a survey in the Akmola region of Kazakhstan. Data were collected from rural households (n = 181) using a face to face questionnaire between 10 August 2019 and 31 October 2019. In addition to the survey, we also use complementary information obtained through semi-structured in-depth interviews in the village Nur-Yessil and focus group meetings in the village Mikhaylovka in August 2017 and 2018, respectively. The Akmola region was selected for data collection since it is included in the food belt around the capital, the main purpose of which is to provide food and goods to the main city of the country. Therefore, the development of food production and agriculture in this region is strategically important for the country (https://aqmola.gov.kz/index.php?mod=news&do=read&rel=1004108&lang=ru). Moreover, the Akmola region offered the opportunity to have both rural households, who are engaged with cooperatives, and those which are not.

The region consists of 17 districts, which in turn are divided into rural districts, which are composed of several villages. The Tselinograd, Arshaly, and Akkol districts were selected out of possible 17 considering their:

- location (i.e., the distance to the main market). These three districts differ in terms of their proximity to their main market with 20 km (Nur-Yessil), 70 km (Mikhaylovka), and 130 km (Yenbek) in Tselinograd, Arshaly, and Akkol districts, respectively.
- population (average population for the Akmola region was 69,444 in January 2019, however, without counting Tselinograd and Burabay districts, which are capital and resort areas respectively, the average for the remaining 15 districts was 23,944 people); and,
- availability of dairy companies in the districts, since under the programme, an increase was expected in milk supplies from rural households (through cooperatives) to dairy companies (LLP "AF Rodina", Production Cooperative "Izhevskiy" and LLP "Eco Milk" are functioning in the selected Tselinograd, Arshaly and Akkol districts respectively); and,
- infrastructure (i.e., good transport interchange).

Thus, selected districts were generally differentiated by distance (Figure 3), and similarity of other factors (e.g., population and dairy factories nearby).

During the selection of rural districts (Table 2), the main criteria were (a) the location (i.e., not alongside the highway, at the same time not very far apart from it); (b) access to broadband (the survey was conducted in person using a tablet but data collected were uploaded online); and (c) the number of rural households (i.e., half or more of them are engaged in farming).

**Table 2.** Descriptive data of the selected rural districts.

| Selected Rural Districts (r.d.) | Number of Villages Included into r.d. | Distance of r.d. from Nur-Sultan (appr.) | Transport Access (Highway) | Households Keeping Livestock and Poultry/Number of Households | Population | Internet (4G) Coverage |
|---|---|---|---|---|---|---|
| 1. Nur-Yessil | 3 | 20 km | Nur-Sultan–Kokshetau | 299/615 | 2800 | yes |
| 2. Mikhaylovka | 3 | 70 km | Nur-Sultan–Karagandy | 413/534 | 1645 | yes |
| 3. Yenbek | 3 | 130 km | Nur-Sultan–Kokshetau | 181/342 | 1053 | yes |

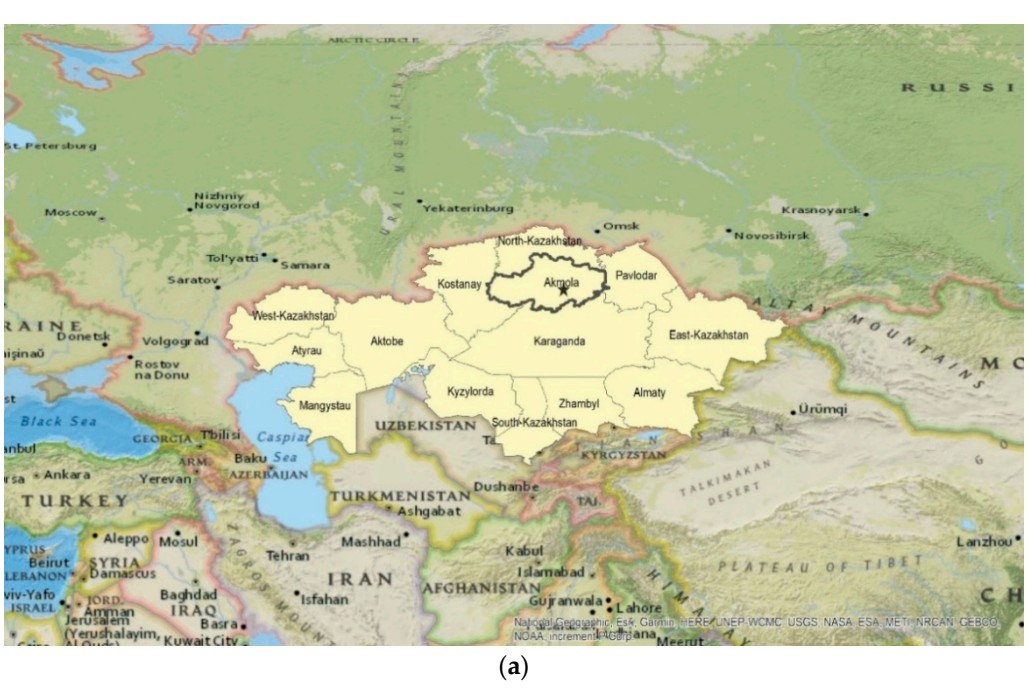

(**a**)

**Figure 3.** *Cont.*

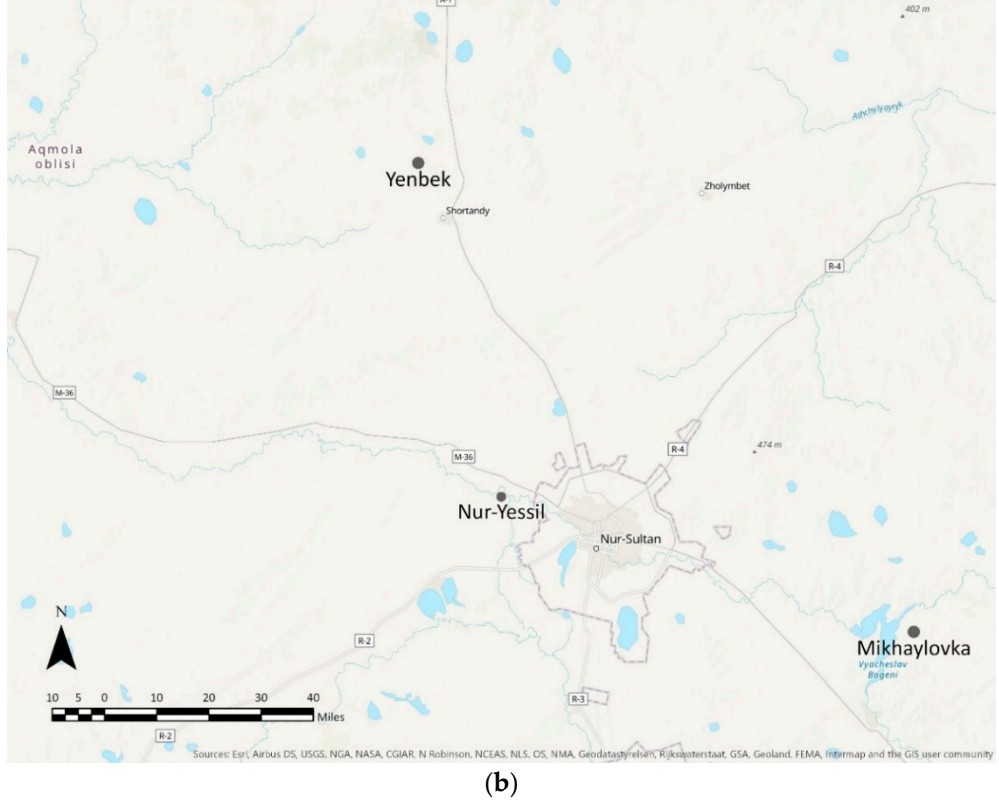

(**b**)

**Figure 3.** The map of the selected rural districts (pictures a and b). (**a**) The map of the Republic of Kazakhstan with a mark of the selected region and the capital, (**b**) Indication of the selected villages by their proximity to Nur-Sultan (the capital).

Data from regional authorities in each r.d. for 1 July 2019. Note: population and number of households are increasing closer to Nur-Sultan.

The main selection criterion to identify respondents was that rural households must own at least a dairy cow. We used the snowball sampling technique by contacting previously identified heads of households in the selected villages to voluntarily take part in the study (i.e., by contacting colleagues and representatives of regional authorities). Participants were also recruited via a semi-random selection process (At each visit to the village for the survey, in addition to meeting with a predetermined respondent, we also knocked on those houses where hay for cattle could be seen, which usually lie on the roof of the barn and in most cases were cow's feeding).

The main instrument used to collect information was a tablet-based questionnaire using Qualtrics software. All participants were provided with an information sheet and consent form containing information about the aims and objectives of the research.

The questionnaire was created in English, then translated to Kazakh and Russian. A second independent person reviewed and edited the translation for accuracy and natural flow of the target language.

The questionnaire consisted of 7 sections (RAA; cultural features; risk attitudes; production and support; information/awareness; socio-demographic; and questions on rural household's intention to join and create a cooperative) and included a total 63 questions and the average duration to complete by the respondents was 30 min.

To reveal psychological aspects underlying rural household's intention to participate in collective action, initially, there were defined salient beliefs of respondents during the pilot study in April 2019 by asking open-ended questions towards the participation in the governmental policy aiming at rural cooperative production, following that, the statements were identified and included into the survey (Table 1).

In order to help capturing cultural aspects, several statements about (a) the rural household's attitude in relation to the former Soviet Union, (b) views towards being self-employed or working in collective, and (c) the level of trust in different groups (i.e., relatives or dairy companies) were included into the survey to identify the impact of them on the behaviour (Table A1 in the Appendix A).

To elicit the risk attitudes of the rural households, statements [47] and self-stated risk aversion [48] methods were used (Table A1 in the Appendix A).

The production and support part of the survey included questions about how much the rural households produce, how much they earn as a dairy farmer, as well as, if they ever received any kind of support from governmental/non-governmental organizations (Table A1 in the Appendix A).

The information/awareness block explored the case of the rural households being previously aware of the policy creating cooperatives along with understanding principles of cooperatives and knowing members of existing cooperatives (Table A1 in the Appendix A).

Socio-demographic set of statements were included questions regarding age, gender, education, and nationality (Table A1 in the Appendix A).

### 2.4. Statistical Analysis

The analysis comprised a combination of methods including principal component analysis (PCA) and parameter model estimation using a bivariate probit model.

### 2.4.1. Principal Component Analysis (PCA)

We used a PCA to avoid a degrees of freedom problem whilst minimising information loss. Hence, PCA is used to reduce the number of variables associated with cultural features, risk attitudes, the production structure, and the level of information. Thus, a set of correlated variables were transformed into a set of independent variables, referred to as components [49,50]. These components were incorporated in the model as explanatory variables.

### 2.4.2. Bivariate Probit Model

The bivariate probit model was used for analysing the factors influencing the rural household's motivation to participate in governmental policy aiming at cooperative production, either by joining or creating a cooperative. Bivariate probit models are used to estimate interrelated decisions. Since individual's intentions to creating and joining a cooperative may be correlated, we used a bivariate probit model to jointly estimate the influence of independent variables on two dependent variables (i.e., joining and creating) allowing for the error terms to be freely correlated. As part of this process, we tested whether such correlation is present. If outcomes are found to be uncorrelated, then two independent probit models can be used instead of the bivariate probit model. As this, A, SN, and PBC statements, generated as independent variables following the given Equations (1)–(3); additional constituents of the model grouped and extracted as components by PCA (e.g., PCA on cultural features, risk attitudes etc.); socio-demographic variables and location are composed explanatory part of the bivariate probit model. Intention to join a cooperative (i.e., would join) and to create a cooperative (i.e., would create) are considered as dependent variables. The formula of the bivariate probit model can be expressed as:

$$
\begin{aligned}
y_1^* &= x_1'\beta_1 + \varepsilon_1, y_1 = 1 \text{ if } y_1^* > 0, \ 0 \text{ otherwise} \\
y_2^* &= x_2'\beta_2 + \varepsilon_2, y_2 = 1 \text{ if } y_2^* > 0, \ 0 \text{ otherwise} \\
&\left( \frac{\varepsilon_1}{\varepsilon_2} \Big| x_1, x_2 \right) \sim N\left[ \begin{pmatrix} 0 \\ 0 \end{pmatrix}, \begin{pmatrix} 1 & \rho \\ \rho & 1 \end{pmatrix} \right],
\end{aligned}
\tag{4}
$$

where $y_1$ and $y_2$ are household's intention to join a cooperative and to create a cooperative, respectively; $x_1$ and $x_2$ represents underlying factors of two types of intention, $\beta_1$, $\beta_2$ are vectors of coefficients and $\varepsilon_1$, $\varepsilon_2$ are the error terms [51].

## 3. Results and Discussions

### 3.1. Descriptive Statistics

Table 3 shows the descriptive statistics of the socio-demographic variables. Nearly 50% belonged at the age band of 31–49, and up to 65% were aged below 50 years old. One third had high education (University). Gender was differentiated equally between male and female. Just over 40% of the respondents were from Mikhaylovka and another a quarter from Yenbek, the rest (about 30%) of the respondents reside in Nur-Yessil. Finally, the number of Kazakh respondents were slightly more than other nationalities (56%).

**Table 3.** Statistical descriptions of the socio-demographic variables.

| | Obs | Mean | S.D. | Min | Max |
|---|---|---|---|---|---|
| Age 50 and older (base category) | | | | | |
| Age 18–30 | 181 | 0.149 | 0.357 | 0 | 1 |
| Age 31–49 | 181 | 0.503 | 0.501 | 0 | 1 |
| Education (1 = University; 0 = otherwise) | 181 | 0.331 | 0.472 | 0 | 1 |
| Nationality (1 = Kazakh; 0 = otherwise) | 181 | 0.564 | 0.497 | 0 | 1 |
| Nur-Yessil (base category) | | | | | |
| Mikhaylovka | 181 | 0.431 | 0.497 | 0 | 1 |
| Yenbek | 181 | 0.249 | 0.433 | 0 | 1 |
| Gender (1 = male; 0 = otherwise) | 181 | 0.497 | 0.501 | 0 | 1 |

### 3.2. Principal Component Analysis

PCA was conducted for reducing the dimensionality and increasing interpretability whilst minimising the loss of information for the following parts of the survey questionnaire: (a) the production and support part, (b) the information/awareness, (c) cultural features statements, and concluded with (d) the risk attitudes section. The principal components obtained were used as covariates in a bivariate probit model. Overall, 9 components (PC) with eigenvalues ≥1 were extracted by the PCA. The varimax rotated component loadings for the original variables are shown in the Tables 4–7.

**Table 4.** The varimax rotated component loadings for the "production and support" variables.

| Statements | PC1 | PC2 |
|---|---|---|
| Is production of milk and/or dairy products your main occupation? | 0.413 | |
| What percentage of your family income comes from the sale of milk and/or dairy products? | 0.551 | |
| What percentage of milk do you leave for own consumption? | −0.495 | |
| How do you evaluate your profit from dairy business? | 0.403 | |
| How many cows do you currently have in total? | | 0.547 |
| How many cows are milked? | | 0.534 |
| What is the average total dairy production of these cows? | | 0.357 |
| Have you ever received any types of support from (non)governmental organisations? | | 0.510 |

**Table 5.** The varimax rotated component loadings for the "information/awareness" variables.

| Statements | PC3 |
|---|---|
| Did you know about the current policy encouraging rural cooperative production? | 0.365 |
| I have received enough information about cooperatives from responsible bodies | 0.452 |
| I understand the principles of cooperatives | 0.514 |
| I agree with the principles of cooperatives | 0.455 |
| I know people who are members of cooperatives | 0.437 |

**Table 6.** The varimax rotated component loadings for the "cultural features" variables.

| Statements | PC4 | PC5 | PC6 | PC7 | PC8 |
|---|---|---|---|---|---|
| (a) individualistic/collectivistic behaviour | | | | | |
| I like to control my business by myself only | 0.583 | | | | |
| I like being my own boss | 0.601 | | | | |
| I like being free to make my own decisions | 0.547 | | | | |
| Working with others makes work more enjoyable | | 0.710 | | | |
| More people—more ideas for development | | 0.703 | | | |
| (b) trust | | | | | |
| I trust dairy companies | | | 0.618 | | |
| I trust merchants | | | 0.660 | | |
| I trust people in general | | | 0.425 | | |
| I trust my neighbours | | | | 0.670 | |
| I trust my relatives | | | | 0.688 | |
| (c) views on past regime | | | | | |
| During the Soviet Union keeping a cow was easier than now | | | | | 0.558 |
| During the Soviet Union keeping a cow was more profitable than now | | | | | 0.586 |
| During the Soviet Union people had more healthy food | | | | | 0.468 |
| The life is better now than in the Soviet Union | | | | | –0.355 |

**Table 7.** The varimax rotated component loadings for the "risk attitudes" variables.

| Statements | PC9 |
|---|---|
| I like trying new things, because I am adventurous | 0.490 |
| I think that every risk is new opportunity to develop my business | 0.447 |
| Please circle your willingness to take a risk in general | 0.562 |
| Please circle your willingness to take a risk in case of investing and borrowing money | 0.421 |

The measure of sample accuracy by the Kaiser–Meyer–Olkin method (KMO) for the "production and support" variables (i.e., PC1 and PC2), was 0.80; for the "information/awareness" (i.e., PC3), was 0.80; for each aspect of the "cultural features", we carried out different PCAs, thus, KMO for the "individualistic/collectivistic behaviour" (i.e., PC4 and PC5) was 0.65; for the "trust", was 0.64; for the "views on past regime", was 0.68; for the "risk attitudes" (i.e., PC9), was 0.63, consequently, showed that the PCA is appropriate (>0.500). Moreover, the Bartlett sphericity test ($p \leq 0.001$), indicates the suitability of PCA as a data reduction technique.

The first component had positive loadings on the production of milk as the main occupation, income from dairy and profit evaluation, as well as a negative sign on own consumption. Therefore, PC1 captured the relevance of dairy production for the rural household as a source of income and as a business. The second component load included aspects such as number of cows, how many cows were milked, their average productivity and the level of support received (governmental/nongovernmental, including financial and non-financial, i.e., training). Therefore, PC2 captures the rural household's capacity to produce.

The third component relates to the "information/awareness" section of the survey. Hence, PC3 was defined as the level of knowledge of policy and cooperatives that rural households have.

Regarding the "cultural features", five components were identified within this section (PC4-PC8). The loading of PC4 included variables associated with individualistic thinking (i.e., preference to be independent in doing business and making decisions (i.e., own bosses), while PC5 captures perceived benefits of collaboration. In the trust subsection, two components were identified—PC6 and PC7, defined as trust in dairy stakeholders (referred to as trust business) and trust in close people, respectively. Finally, PC8 includes statements that indicate positive views on the Soviet Union (SU) period by rural households as well as the statement "The life is better now than in the Soviet Union." Consequently, PC8 was identified as a positive attitude towards the Soviet Union (i.e., SU nostalgic).

All variables in the "risk attitudes" section had a positive sign, therefore PC9 was intended for risk-seeking rural households.

### 3.3. Bivariate Probit Model

In order to test the appropriateness of the Bivariate model, we tested for the correlation between error terms using a likelihood ratio test. The parameter that measures the correlation between error terms (rho) was found to be statistically different from zero (see LR test in Table 8).

**Table 8.** Results of the bivariate probit model.

| | Would Join | | Would Create | |
|---|---|---|---|---|
| | Coeff. | Z Statistics | Coeff. | Z Statistics |
| Attitude (A) | 0.059 *** | 4.61 | 0.041 *** | 3.23 |
| Social norms (SN) | 0.021 * | 1.64 | 0.016 | 1.30 |
| Perceived behavioural control (PBC) | −0.016 | −0.94 | −0.007 | −0.40 |
| Dairy as a source of income (PC1) | −0.179 * | −1.85 | −0.205 ** | −2.06 |
| Capacity to produce (PC2) | 0.193 ** | 2.00 | 0.140 | 1.50 |
| Awareness and knowledge (PC3) | 0.044 | 0.54 | 0.263 *** | 3.23 |
| Own boss (PC4) | 0.023 | 0.25 | 0.063 | 0.69 |
| Benefits collaboration (PC5) | 0.058 | 0.47 | 0.008 | 0.07 |
| Trust business (PC6) | −0.152 | −1.39 | −0.046 | −0.43 |
| Trust close ones (PC7) | 0.188 | 1.60 | −0.128 | −1.17 |
| SU nostalgic (PC8) | −0.084 | −0.83 | −0.015 | −0.16 |
| Risk (PC9) | 0.241 ** | 2.21 | 0.175 * | 1.69 |
| Age 50 and older (base category) | | | | |
| Age 18–30 | 0.145 | 0.32 | 0.560 | 1.26 |
| Age 31–49 | 0.144 | 0.48 | 0.339 | 1.15 |
| Nur-Yessil (base category) | | | | |
| Mikhaylovka | −0.324 | −0.94 | −0.483 | −1.41 |
| Yenbek | −0.227 | −0.61 | −0.269 | −0.72 |
| Nationality (1. Kazakh) | −0.181 | −0.59 | 0.589 ** | 2.00 |
| Gender (1. Male) | 0.463 * | 1.70 | −0.172 | −0.68 |
| Education (1. University) | −0.018 | −0.07 | −0.524 * | −1.86 |
| Constant | −1.853 *** | −3.64 | −1.524 *** | −3.11 |
| rho (ρ) | 0.647 | 5.56 | | |
| Number of observations | 181 | | | |
| Log-likelihood | −136.656 | | | |
| LR test of rho = 0: chi2(1) = 17.6369 | Prob > chi2 = 0.000 | | | |

Note: *, **, *** for 10, 5, and 1% of significance level, respectively.

The results of the bivariate probit model are presented in Table 8 and show that attitude (A) on intention is positively associated to both behaviours, joining and creating a cooperative (*p*-value < 0.01). Hence, an increase in positive attitudes towards being a member of a cooperative will increase the chance of rural households joining and creating a cooperative. A similar positive association between smallholders' attitudes towards cooperatives and their intention to join a cooperative was found for Romanian smallholders [17]. Farmers' attitude was also found to be related to their intention to uptake rural development policy in North Macedonia, Serbia, and Bosnia and Herzegovina [52]. Regarding social norms, we found a statistically significant association between SN and joining a cooperative (*p*-value = 0.10). More specifically, the greater the respondent's perceived belief has, that others want the rural household to join a cooperative as well as the higher the motivation to comply with them is, the more likely that the respondent will join a cooperative. In other words, "the stronger the perceived social pressure, the more likely it is that an intention to perform the behaviour will be formed" [18]. Previous research on the association between SN and joining and creating cooperative is inconclusive. Whereas Hyland et al., Warsame and Ireri [29,33] found no association between SN and being a member

of a cooperative, Mollers et al. and Sabates-Wheeler [17,53] found that the role of relatives and friends is positively associated with being a member of a cooperative. Finally, we found that PBC (i.e., the extent to which households believe that they are capable of joining and creating a cooperative) was associated with neither joining nor creating a cooperative.

Previous studies on dairy farmers' self-reported value of cooperative membership showed that farmers may benefit from cooperatives in a way of (a) having a stable market channel, (b) low transaction cost, since a cooperative obligated to collect agricultural products, and (c) competitive producer price [54]. The results of this study revealed that structures currently dedicated to sell their produce to the market with clear business orientation (i.e., dairy as a source of income) are less likely to join (*p*-value < 0.10) and create (*p*-value < 0.05) a cooperative than those which do not. It is worth noting that prices received by households may differ depending on the buyer (cooperatives, dairy factories, and direct consumers) due to the structure of the milk market (e.g., market shares of cooperatives in relation to other milk buyers). Markets structure was highlighted previously as a determinant of product price differences amongst other parameters [55,56]. This was found in the discussions with households during the in-depth interviews, which showed that the price offered by dairy factories (following the programme, rural households have to pass the milk to dairy factories through co-operatives, consequently, dairy factories will release the payment for the received milk. Additionally, co-operatives will receive 10 per cent subsidy from the government) is less compared with the direct sales to consumers. Although the existence of cooperatives is argued to lead to higher prices for farmers compared with no cooperatives [57], this study showed that business-orientated rural households (e.g., who might sell to an established market that offers a higher price for a litre of milk) would not be as motivated to participate in the governmental program as those without a business orientation. However, rural households that have a capacity to produce and may need further support (e.g., subsidy, training) are more likely to join (*p*-value < 0.05) a cooperative. For instance, an increase in the number of cows might require more resources, such as feeding, consequently, increase the need in support (e.g., cheapened feeding). These results are somewhat consistent with Jitmun et al. [36] where an increase in herd size is suggested to be positively associated with an increase in the likelihood of rural households to become a member of a cooperative. Wossen et al. [58] have also found that households that have more livestock are more likely to be a member of a cooperative.

As indicated in Table 8, the rural household's intention to create a cooperative is positively associated with being aware of the policy and with having adequate information about cooperatives (*p*-value < 0.01). Discussions with rural households during the focus-group meeting also revealed the interest of participants in creating a cooperative, but most of them noted a lack of information on the procedure and principles for creating cooperatives. This result is in line with the findings of Gong et al. [42] and other research on technology adoption and agricultural insurance purchases by farmers [39,41] which highlights limited knowledge as a factor associated with farmers' unwillingness to join cooperatives. Lerman [59] found that the lack of existing cooperatives and the lack of information about cooperatives were the main two factors for farmers in Kyrgyzstan, a neighbouring country to Kazakhstan, for not being a member of a cooperative.

Interestingly, we did not find any correlation of cultural features [15,35,38,46,60] on the rural household's motivation to create/join a cooperative, despite expecting these to be essential determinants of rural household's participation in collective actions. Additionally, no difference was found between the three locations covered.

Regarding risk perception, we investigated whether household's attitudes towards risk may be associated with their decision to participate in collective actions (i.e., join/create a cooperative). We found that risk seeking rural households are more likely to join (*p*-value < 0.05) and create (*p*-value < 0.10) a cooperative. Results indicate that rural households that are willing to take risks in general and/or take risks in case of investing and borrowing money; households that like trying new things; and/or take every risk as a new opportunity to develop their business will be more likely to join/create a cooperative. Although one could expect that more risk-averse individuals would be more likely to

join/create co-operatives since these are a form of risk management [39], it is possible that risk-seeking behaviour is the result of risk-averse motives (e.g., avoiding production risks) by rural households [61].

Finally, the socio-demographic characteristics of rural households were found to be associated with both joining and creating a cooperative. Men are more likely to join a cooperative than women (*p*-value < 0.10). This result is somewhat consistent with the study of Gebre et al. [62] on agricultural technology adoption which also highlighted the positively significant role of being male headship in the decision making. Ahmed and Mesfin [63] have also indicated men are more likely to participate in agricultural cooperatives than women. At some point, it can be an implication of the socio-cultural norms, which makes great discrimination in women's ability to make decisions [64,65] or just because women are less considered on farming in a long scale perspective [66].

Although a number of studies found a positive correlation between education and willingness to participate in collective actions [36,39,63], the results of this study showed that higher educated rural households are less likely to create a cooperative (*p*-value < 0.10). It might be due to the agriculture in Kazakhstan being unattractive from a social viewpoint, therefore, higher educated rural households are seeking more prestigious and well-paid professions to preserve their social status [67,68]. Age of rural households was found not being correlated with their intention to join/create a cooperative, in spite of it having a positive and significant relationship with cooperative membership in previous studies [63,69]. In addition, Kazakh (*p*-value < 0.05) are more likely to create a cooperative than other nationalities, which can be explained by the Kazakh nation being an indigenous nation in Kazakhstan, and compared with other ethnic groups, the Kazakh desire more to manage social processes in the country, i.e., being active in political-economic decision-making [70].

Additionally, we have run a set of model specifications including a bivariate probit model using RAA variables only, a bivariate probit model using components derived from the PCA, and a bivariate probit model with only significant covariates, that can be found in the Supplementary Materials. We conducted LR tests to compare the different model specifications. The full model (Model 1) was found to perform better than models using only reason action approach variables (Model 2) and only components derived from PCA (Model 3). No difference was found between Model 1 and model using significant variables only (Model 4). However, we present results for Model 1 to show the significant level of all independent variables used in the analysis.

## 4. Conclusions

The dairy sector of Kazakhstan is experiencing structural problems, such as the prevail of rural households and their disconnection with the supply chain, that leads to low milk productivity of the sector. Co-operatives are suggested by the government to be a way to increase milk supply to dairy factories via creating a formal network of producers (i.e., rural households). We analysed the factors underlying rural household's motivation to join or create a cooperative in Kazakhstan. A number of policy recommendations can be proposed on the basis of this research.

Firstly, the study has highlighted the importance of psychological factors such as holding positive attitudes toward cooperative production and perceived social norms on the decision to join and create a cooperative. More specifically, an increase of beliefs on benefits of cooperative production as well as support from social referents would increase the chance of the governmental programme to be accepted by rural households. Moreover, the results showed that being aware of the cooperative production-related policy and having adequate knowledge of concepts of cooperatives would increase the chance of rural households to create a cooperative. Additionally, we found that risk-seeking rural households are more inclined to join and create a cooperative.

This indicates that policies aiming at increasing awareness of the benefits of cooperatives by providing information on cooperative creation would be advisable. More specifically, our findings suggest that such policies should target not only individual rural households but communities as a whole too, where social referents can also be informed and influence rural households' decisions. This provision of information could be deployed by extension services on the basis of currently existing

organisations, such as Atameken. This would help rural households and communities to understand the main differences between production systems and, in addition, inform rural households about the benefits of joining and creating a cooperative and avoid uncertainty. Extension services work could be complemented using mass media and the regional authorities to extend the impact of informing rural households on cooperatives.

To better target households and communities, policies should take into account the current business orientation of rural households in producing milk. Thus, emphasis should be put on rural households in which dairy is not currently a main source of income, but they have the capacity to increase milk production. Thus, for supporting the increase in milk productivity through the rural household's (either joining or creating a cooperative), supportive policies (e.g., subsidies to increase capacity, training to increase awareness) are recommended for rural households that have the capacity to increase production. Conversely, rural households which are business oriented can be attracted by the guarantee of sales or increase in the price of milk.

Additionally, gender and nationality are significantly correlated with joining and creating, respectively. These results suggest that not all member of society may have the same interest and/or opportunities. We recommend a policy to be inclusive to ensure support to both genders as well as all nationalities.

To conclude, this study provides guidelines and suggestions for policy makers and stakeholders of the agricultural sector. We offer key information to consider when preparing documents to successfully create and support agricultural cooperatives in Kazakhstan. Furthermore, findings presented in this paper might also be relevant for post-communist countries, where small-scale agriculture prevails.

**Supplementary Materials:** The following are available online at http://www.mdpi.com/2077-0472/10/11/568/s1, Table S1: The results of the Bivariate Probit Model using RAA variables only; Table S2: The results of the Bivariate Probit Model using components derived from the PCA; Table S3: The Bivariate Probit Model with only significant covariates; Table S4: Comparison of the results from different models; Table S5: Correlation matrix for RAA variables and components (PCs) derived from the PCA.

**Author Contributions:** Conceptualization, S.K., F.J.A. and Y.G.; Methodology, S.K., F.J.A. and Y.G.; Software, S.K., F.J.A. and Y.G.; Formal Analysis, S.K., F.J.A. and Y.G.; Investigation, S.K., F.J.A. and Y.G.; Data Curation, S.K.; Writing—Original Draft Preparation, S.K.; Writing—Review & Editing, F.J.A. and Y.G.; Visualization, S.K., F.J.A. and Y.G.; Supervision, F.J.A. and Y.G. All authors have read and agreed to the final version of the manuscript.

**Funding:** This study was funded by the University of Reading and the Centre for International Programs (CIP) "Bolashak" of the Republic of Kazakhstan.

**Conflicts of Interest:** The authors declare no conflict of interest.

## Appendix A

**Table A1.** All other questions included in the survey.

| Questionnaire Statements | Likert Scale from 1 = Strongly Disagree to 5 = Strongly Agree |
|---|---|
| Production and Support | |
| How many cows do you currently have in total? | stated numbers |
| How many cows are milked? | stated numbers |
| What is the average total dairy production of these cows? | stated numbers |
| Have you ever received any types of support from (non)governmental organisations? | yes/no |
| Is production of milk and/or dairy products your main occupation? | yes/no |
| What percentage of your family income comes from the sale of milk and/or dairy products? | 0; 1–25%; 26–50%; 51–75%; 76–100% |
| What percentage of milk do you leave for own consumption? | stated numbers (percentage) |
| How do you evaluate your profit from dairy business? | profit > expenses profit = expenses profit < expenses |
| Information/Awareness | |
| Did you know about the current policy encouraging rural cooperative production? | yes/no |
| I have received enough information about cooperatives from responsible bodies | Strongly disagree–strongly agree |
| I understand the principles of cooperatives | Strongly disagree–strongly agree |
| I agree with the principles of cooperatives | Strongly disagree–strongly agree |
| I know people who are members of cooperatives | Strongly disagree–strongly agree |
| Cultural Features | |
| I like to control my business by myself only | Strongly disagree–strongly agree |
| I like being my own boss | Strongly disagree–strongly agree |
| I like being free to make my own decisions | Strongly disagree–strongly agree |
| Working with others makes work more enjoyable | Strongly disagree–strongly agree |
| More people-more ideas for development | Strongly disagree–strongly agree |
| I trust my neighbours | Strongly disagree–strongly agree |
| I trust my relatives | Strongly disagree–strongly agree |
| I trust dairy companies | Strongly disagree–strongly agree |
| I trust merchants | Strongly disagree–strongly agree |
| I trust people in general | Strongly disagree–strongly agree |
| During the Soviet Union keeping a cow was easier than now | Strongly disagree–strongly agree |
| During the Soviet Union keeping a cow was more profitable than now | Strongly disagree–strongly agree |
| During the Soviet Union people had more healthy food | Strongly disagree–strongly agree |
| The life is better now than in the Soviet Union | Strongly disagree–strongly agree |
| Risk Attitude | |
| I like trying new things, because I am adventurous | Strongly disagree–strongly agree |
| I don't like changes in my life | Strongly disagree–strongly agree |
| I think that every risk is new opportunity to develop my business | Strongly disagree–strongly agree |
| Please circle your willingness to take a risk in general | from 1 to 5 |
| Please circle your willingness to take a risk in case of investing and borrowing money | from 1 to 5 |
| Socio-Demographic | |
| Age | 18–30; 31–49; 50 and older |
| Education | = 1 if University = 0 otherwise |
| Gender | = 1 if Male = 0 if otherwise |
| Nationality | = 1 if Kazakh = 0 otherwise |

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
