# Peer review of "Attitudes of Kazakh Rural Households towards Joining and Creating Cooperatives"

_agriculture, doi:10.3390/agriculture10110568_

Round 1
Reviewer 1 Report
Dear authors,
thank you very much for these studies on current stage of agricultural production and national policies in Central Asia. I generally have only few comments:
- variables in your model can be specified instead of using abbreviations (at least I would try to do so)
- map of study site could be improved
- in the discussion part I found less studies from the former USSR, those from Nepal or China are good, but it is always nice to compare the results with similar countries, and, this could also increase generally available literature on former USSR (which we still lack I'd say)
Thank you for your work.
Best Regards,
Author Response
Dear Reviewer,
We would like to thank you for your useful comments. We believe the manuscript has benefited comments and suggestions made by reviewers.
RESPONSE TO COMMENTS TO REVIEWER #2
- variables in your model can be specified instead of using abbreviations (at least I would try to do so)
We have defined the variables in the revised manuscript. Line 394-395 (Table 7)
- map of study site could be improved
We have now inserted an updated map in the revised manuscript
This new map has higher resolution and analysis (96 dpi) and bit depth (24).
- in the discussion part I found less studies from the former USSR, those from Nepal or China are good, but it is always nice to compare the results with similar countries, and, this could also increase generally available literature on former USSR (which we still lack I'd say)
We have extended our search following recommendation from reviewer #2. Unfortunately, we have not been able to find much literature linked to more similar countries to Kazakhstan (former USSR members), apart from a study in Kyrgyzstan and the literature already included in the manuscript referring to rural development policies in former communist regimes such as Rumania, Republic of North Macedonia, Serbia, Bosnia and Herzegovina.
We have added the following text to the manuscript in order to further enhance the arguments in the discussion part LINES 427-429: “Lerman (2013) found that the lack of existing cooperatives and the lack of information about cooperatives as the main two factors for farmers in Kyrgyzstan, a neighbouring country to Kazakhstan, not being a member of a cooperative.”
Reviewer 2 Report
Comments/Suggestions:
- (LINES 2-3) "Attitudes of Kazakh rural households towards cooperative production."
Your description of the co-operatives is unclear. Exactly what is the situation? The co-operatives are formed by individual households who then supply milk? Especially considering the history of the Soviet Union, it is not clear if you mean “collective marketing/processing” or “collective production/agriculture”.
- (LINE 286)
You are not conducting a bivariate probit model, which concerns the relationship of ONE outcome variable (y1) to ONE covariate (x1). Instead, you have TWO multivariate probit models (y1, y2) and MULTIPLE covariates (x1 ... xn).
- (LINES 48-49) “b) small/peasant farms and c) rural households.”
What is the difference between small farms and rural households?
- (LINES 56-59) “Although rural households are the main national producers of milk, they are considered as a subject to informal trade (i.e. direct sales to consumers). This system leads to dairy factories having a deficit of milk processing, and consequently, a low level of processed dairy products in the country (Sheikin & Kulbayeva, 2015).”
Why can rural households not deliver directly to dairy factories (processors)? Transaction cost? Law?
- (LINES 60-62) “The situation turned out in such a way that the government considered measures to increase milk processing capacity in Kazakhstan and introduced various policies to stimulate milk production at an industrial production level”.
But according to the previous paragraph, the problem is the inability of rural households to deliver milk to processors, who then experienced excess capacity. Something is not right...
- (LINES 79-80) “However, the determinants behind rural households’ motivation to create and/or join a cooperative are not well understood yet.”
Perhaps in the context of Kazakhstan. Elsewhere, there is a huge amount of literature on the various determinants of co-operative membership and formation. Such models (co-op membership = f(member characteristics)) are often estimated in the first stage of a two-stage regression with some type of output as the outcome variable in the second stage. The studies are too many to list here, but you can look at Grashuis and Su (2019) for a review.
- (LINES 146-162)
I do not fully understand the scales. (a) Why do you use two different types of scales (i.e. from 1 to 5, from -2 to 2)? Do you not think respondents may have become confused? (b) How do you arrive at the range of -60 to 60? I see six attitudes, each measured in terms of strength and evaluation. So 12 items with a maximum score of 5. So should the range not be 0 to 60?
- (LINES 185-187) “Therefore, it is expected that rural household’s decision to join or create a cooperative depends on their risk attitudes, with risk averse rural households being less likely to make changes to the status-quo.”
Co-operative membership is sometimes conceptualised as risk management. Co-operatives create access to input and output markets, which implies risk reduction on the part of the members.
- (LINES 209-211) “Moreover, Akmola region offered the opportunity to have both rural households who are engaged with cooperatives and those which are not.”
I am worried about how prior and current experience with co-operatives is measured and included in the model. It appears to relate to the “awareness/information” factor, which has a significant relationship to the second outcome variable (would create co-operative). However, it is not entirely clear if past/current membership is in fact measured and included. If not, omitted variable bias is likely present.
- (LINES 316-320)
(a) If possible, try to merge Tables 2-6. (b) Can you report the correlation coefficients for A, SN, PBC, and the nine PCs?
- (LINES 242-244) “The main instrument used to collect information was a tablet-based questionnaire survey by using Qualtrics software. All participants were provided with an information sheet and consent form containing information about the aims and objectives of the research.”
So to be clear, you contacted the respondents in person, and then administered the survey on Qualtrics? If you are physically present at the time of the response, is it then still an online survey? I think not.
- (LINES 204-206) “In addition to the survey, we also use complementary information obtained through semi structured in-depth interviews in the village Nur-Yessil and focus group meetings in the village Mikhaylovka in August 2017 and 2018, respectively.”
What is the nature of the complementary information? How is the information used in your analysis?
- (LINES 202-204) “Data was collected from rural households (n=185) using a face to face questionnaire between August 10th, 2019 and October 31st 2019.”
I miss summary statistics of the demographic characteristics of your respondents. What information do you have? Age, gender, education, income, location, household size, farm size? You must present a profile of your respondents.
- (LINE 360)
(a) You will need to try and report more models. Maybe with only A, SN, and PBC. Only the nine PCs. Only the significant covariates. (b) What are the base categories for some of your control variables?
- (LINES 364-366) “The discussions with households during the in-depth interviews showed that the price offered by cooperatives is less compared with the direct sell.”
Generally, in this section, you must do much better in terms of comparing and contrasting your results to prior results. Here, for example, you miss an opportunity to cite Carletti et al. (2018) and Grashuis (2020), who both observed how co-operatives offer relatively low prices BUT for a reason. Usually, the difference is more than compensated by end-of-the-year refunds and dividends. The same reason may apply in your case.
- (LINES 387-388) “Additionally, no difference was found between the three locations covered.”
Why would you even expect regional heterogeneity? Are there co-operatives in one region but not the other two?
- (LINES 392-393) “Finally, the socio-demographic characteristics of rural households were found to be associated with both joining and creating a cooperative.”
This is why it is so important to share the demographic characteristics of your survey respondents. How many of your respondents are female? If your survey is like most other farm surveys, then there will be few female respondents. If so, the ability of your model to accurately estimate a relationship of gender to the willingness to join/create a co-operative is in doubt.
18.
The works of Alho (2015; 2016; 2017) should probably be referenced. Alho’s work relates directly to your study.
- (LINES 404-406) “It might be due to the agriculture in Kazakhstan being unattractive from a social viewpoint, therefore, higher educated rural households are seeking more prestigious and well-paid professions to preserve their social status (BednaÅ™íková et al., 2016; Otar et al., 2020).”
I see a potential problem. Are rural households not farms? Are some of your respondents not farms? Are you asking non-farm respondents if they are willing to join/create farmer co-operatives?
- (LINES 440-443) “Summing up, this study provides guidelines and suggestions for policy makers to consider when preparing documents to create and support agricultural cooperatives in Kazakhstan, however, the findings presented in this paper might also be interesting for post-Soviet countries, where small scale agriculture prevails.”
This section is just an ordinary summary. You do not draw conclusions. You say the study provides guidelines for policymakers. So what are these guidelines? What are the implications of your findings for policymakers? And what about farmers, co-operatives, and other stakeholders? A policymaker is likely to only read the abstract/conclusion of your study; these are the most important parts of your manuscript.
Author Response
Dear Reviewer,
We would like to thank you for your useful comments. We believe the manuscript has benefited comments and suggestions made by reviewers. Please see the attachment
Kind regards,

Round 2
Reviewer 2 Report
First of all, thank you for the fast and thorough response to the first round of comments.
Where possible, you appear to have addressed my various concerns and recommendations. Furthermore, you have educated me in terms of the correct interpretation of bivariate probit models. I am pleased with the state of the revised manuscript, but I still have minor recommendations for you to consider:
- In terms of risk, because co-operatives are a form of risk management, you would expect more risk-averse individuals to be more likely to join/create co-operatives. In lines 489-491 you discuss the result, but the relationship is not entirely clear. Who are more likely to join/create co-operatives, risk-averse or risk-seeking individuals?
- You say the disturbances of the two models are correlated. Do you actually have a statistic as proof? Otherwise it is unclear why you are conducting the bivariate model.
- Report Table A2 in the main body of the manuscript, not in the appendix. Also briefly discuss the sample characteristics in the data section. It really is important information.
- In lines 517-519 you refer to the extra models. These often facilitate a "robustness check" or "sensitivity analysis", if done properly. Can you compare the results of these models to your main models and see any differences? If so, they should be discussed in the text.
- The language is understandable, but the manuscript would still benefit from some editing by a native speaker.
Good luck with your research.
